

# Growth response of *Emiliania huxleyi* to ocean alkalinity enhancement

Giulia Faucher[1], Mathias Haunost[1], Allanah Paul[1], Anne Ulrike Christiane Tietz[1], Ulf Riebesell[1]

[1]Biological Oceanography, GEOMAR Helmholtz Centre for Ocean Research Kiel, 24148, Kiel, Germany

*Correspondence to*: Giulia Faucher (gfaucher@geomar.de)

**Abstract.**

The urgent necessity of reducing greenhouse gas emissions is coupled with a pressing need for widespread implementation of carbon dioxide removal (CDR) techniques to limit the increase in mean global temperature to levels below 2°C compared to pre-industrial times. One proposed CDR method, Ocean Alkalinity Enhancement (OAE), mimics natural rock weathering

processes by introducing suitable minerals into the ocean thereby increasing ocean alkalinity and promoting $CO_2$ chemical absorption. While theoretical studies hold promise for OAE as a climate mitigation strategy, careful consideration of its ecological implications is essential. Indeed, the ecological impacts of enhanced alkalinity on marine organisms remain a subject of investigation as they may lead to changes in species composition. OAE implicates favourable conditions for calcifying organisms by enhancing the saturation state of calcium carbonate and decreasing the energetic costs for calcification. This may

affect marine primary production by improving conditions for calcifying phytoplankton, among which coccolithophores play the leading role. They contribute <10% to the global marine primary production, but are responsible for a large proportion of the marine calcite deposition. While previous research has extensively studied the effects of ocean acidification on coccolithophores, fewer studies have explored the impacts of elevated pH and alkalinity. In this context, we studied the sensitivity of *Emiliania huxleyi*, the most widespread coccolithophore species, to ocean alkalinity enhancement in a culture

experiment. We monitored the species' growth and calcification response to progressively increasing levels of total alkalinity (TA). Above a change in total alkalinity (ΔTA) of ~ 600 µmol kg$^{-1}$, as $CO_2$ concentrations decreased, *E. huxleyi* growth rate diminished, suggesting a threshold $CO_2$ concentration of ~ 100 µatm necessary for optimal growth. The cellular calcite to organic carbon ratio (PIC:POC) remained stable over the total alkalinity range. Due to the decreasing growth rate in response to alkalinity enhancement, total carbonate formation was lower. OAE is rapidly advancing and has already reached the field-

testing stage. Hence, our study contributes to the most critical part of investigations required to comprehend potential biological implications before large-scale OAE will be adapted.

## 1 Introduction

Governments have recognized the need to constrain human-induced climate change and declared to keep the mean global temperature rise well below 2°C compared to pre-industrial values to prevent dangerous consequences associated with a further



increase in temperature (United Nations Framework Convention on Climate Change, 2015). Currently, the rate at which greenhouse gas emissions are being cut is slow and insufficient to meet this target, therefore the use of negative emission technologies (NET) to remove atmospheric $CO_2$ is expected to be inevitable in supporting efforts to limit global temperature rise  (IPCC, 2021). One proposed method to remove $CO_2$ from the atmosphere mimics the natural process of rock weathering, whereby suitable minerals are extracted and introduced into the surface ocean (Ocean Alkalinity Enhancement, OAE; Gattuso

et al., 2015; GESAMP, 2019). The dissolution of the respective minerals consumes protons and conservative cations are released, which leads to a shift of the carbonate chemistry species from $CO_2$ towards $HCO_3^-$ and $CO_3^{2-}$, thus to an increase in alkalinity (Wolf-Gladrow et al., 2007; Bach et al., 2019). As a result, the seawater becomes undersaturated with $CO_2$, which is then balanced by additional absorption of atmospheric $CO_2$. Hence through OAE, $CO_2$ is chemically absorbed and subsequently safely stored mainly as bicarbonate over geological timescales (e.g. Hartmann et al., 2013).

Theoretical studies indicate that OAE has the potential for large scale application to absorb gigatonnes of carbon from the atmosphere and thus contribute to global climate change mitigation efforts (e.g. Keller et al., 2014;  Renforth and Henderson, 2017; Taylor et al., 2016). This provides a basis for testing the applicability, efficiency and ecological impacts of this approach in practice. Key factors in this context are the impacts of the seawater chemistry changes arising from enhanced alkalinity on primary production and biogenic calcification. In a recent paper Bednaršek et al. (2024), offered a conceptual synthesis of the

responses of marine calcifiers to OAEs, utilizing insights from multiple ocean acidification (OA) studies. This study evidenced that only 40% of the analysed groups show a neutral response upon alkalinity addition. For marine microbes that rely on dissolved carbon to fuel primary production the shifts in seawater carbonate chemistry are relevant for their physiological processes. Below certain thresholds, dissolved $CO_2$ concentrations can indeed limit phytoplankton growth and lead to reduced primary production (Riebesell et al., 1993). In addition, OAE leads to beneficial changes in carbonate chemistry for biogenic

calcification due to an increased pH and carbonate saturation state and thus may promote calcifying organisms (Bach et al., 2019; Bednaršek et al., 2024). The formation of calcium carbonate, however, consumes alkalinity and is a $CO_2$ source (Zeebe and Wolf-Gladrow, 2001), so increased calcification rates would counteract $CO_2$ drawdown through OAE.

Coccolithophores are the most important calcifiers in marine phytoplankton. They contribute significantly to the marine primary production in the surface ocean and the deposition of calcite in deep-sea sediments (Poulton et al., 2007; Broecker

and Clark, 2009). In the last decades, many studies investigated the impact of OA on the growth and calcification of coccolithophores (e.g. Riebesell et al., 2000 Langer et al., 2006; Bach et al., 2013; Faucher et al., 2020; Paul and Bach, 2020) and evidenced the sensitivity of these organisms to changes in carbonate chemistry. Increased $p$CO_2 and lower pH induce significant effects on the growth and calcification of coccolithophores (e.g. Riebesell et al., 2000; Beaufort et al., 2011; Bach et al., 2011, 2015). Most efforts focused on manipulating the level of $p$CO_2 rather than alkalinity. Only a few of them

investigated the effect of higher pH and lower $p$CO_2 on these organisms (e.g. Bach et al., 2015; Langer et al., 2006) indicating a reduction in the growth rate of the species tested when $p$CO_2 is below ∼ 100 µatm (Sett et al., 2014; Bach et al., 2011; 2015) and offered first insights into the potential effects of OAE on coccolithophore growth.



OAE is based on adding alkalinity through different approaches, which raises carbonate chemistry parameters in various ways. Recently, Gately et al. (2023) showed a neutral response of two phytoplankton species, the coccolithophore *Emiliania huxleyi*

and the diatom *Chatoceros* sp. to OAE in an experiment where the culture media were bubbled for some days to equilibrate with the atmosphere before the algae were inoculated. This approach avoided a strong perturbation of the carbonate chemistry (i.e. avoidance of pH spikes and $p$CO$_2$ limiting conditions). Their results show that even in the highest alkalinity treatment ($\Delta$TA ~ 2700 µmol kg$^{-1}$), the $p$CO$_2$ was above 300 $\mu$atm and the pH only slightly increased by 0.4 unit. The apparent resilience of these species to increased alkalinity in the study from Gately et al. (2023) is encouraging considering a potential application

of OAE. However, in a non-equilibrated OAE approach where atmospheric equilibration of CO$_2$ is prevented (*sensu* Hartmann et al., 2022; Suitner et al., 2023), the carbonate chemistry shifts towards lower $p$CO$_2$ and higher [CO$_3^{2-}$] and pH. The impact of these more extreme conditions on marine calcifiers, as coccolithophores, has not yet been sufficiently investigated in order to assess the effects of ocean OAE on marine primary production.

*Emiliania huxleyi* is the most abundant and widespread coccolithophore species in the modern ocean (Westbroek et al., 1993)

and served as a model species in plenty of studies that investigated the impact of ocean acidification on coccolithophores (see Wheeler et al., 2023). Bednaršek et al. (2024) concluded that *E. huxleyi* stood out as the sole coccolithophore species not displaying a neutral effect of increased alkalinity on calcification.

In this study, we tested the response of *E. huxleyi* in growth and calcification under progressively increasing total alkalinity (TA) levels. The increase in alkalinity was achieved by adding sodium hydroxide (NaOH) solution to the culture medium. The

medium was not allowed to equilibrate with the atmosphere after NaOH addition and the response of *E. huxleyi* to decreasing CO$_2$ concentrations and increasing pH due to enhanced alkalinity was investigated. The study aimed to test whether i) the growth rate of *E. huxleyi* decreases under reduced CO$_2$ concentrations due to enhanced alkalinity and ii) the carbonate chemistry conditions under increasing TA (i.e. increasing [HCO$_3^-$], lower [H$^+$]) that are conducive to calcification, lead to physiological disbalances between calcification and growth.

**2 Materials and Methods**

**2.1 Culture conditions and experimental design**

Monospecific cultures of *Emiliania huxleyi* (B92/11) were grown in batch cultures at low biomass (< 50000 cell mL$^{-1}$) to avoid a significant impact of biological processes on the chemical conditions of the growth medium (LaRoche et al., 2010). *Emiliania huxleyi* was grown at 15°C in a photoperiod of 16:8 hours light to dark and a photon flux of 150 µmol m$^{-2}$ s$^{-1}$ (measured with

a Li-Cor, HeinzWalz GmbH, Effeltrich). The growth medium was based on an artificial seawater (Kester et al., 1967) with a salinity of 34, which was enriched with 64 µmol kg$^{-1}$ NaNO$_3$, 4 µmol kg$^{-1}$ NaHPO$_4$, and trace metals and vitamins according to the f/8 medium (Guillard and Ryther, 1962). In addition, 10 nmol kg$^{-1}$ SeO$_2$ was added (Danbara and Shiraiwa, 1999) as well as 2 mL kg$^{-1}$ of 0.2 µm filtered natural seawater to prevent potential growth limitation of *E. huxleyi* due to other substances not included in the f/8 receipt (Bach et al., 2011). The medium was filtered (0.2 µm) into sterile 0.6 L polycarbonate bottles



where different amounts of NaOH (1M) solution were added to set up an alkalinity gradient among the replicates. The alkalinity gradient was set up across 16 distinct treatments, with intervals of 50-100 µmol kg$^{-1}$ from ambient TA levels of 2350, up to ~3500 µmol kg$^{-1}$. After the addition of NaOH, the bottles were sealed gastight and the headspace inside was kept as low as possible to minimize gas exchange. Small volumes of a pre-culture with exponentially growing *E. huxleyi* were pipetted into the bottles to meet target cell concentrations of around 100 cells mL$^{-1}$. The cells were acclimated for 7-9 generations to the

respective alkalinity conditions and samples of 0.5 mL were taken at selected time points to monitor the concentration and growth of *E. huxleyi*. After the acclimation phase, a small volume of the cultures was transferred into a further set of polycarbonate bottles (2.7 L) filled with fresh NaOH-manipulated medium. Cells were homogenized by gentle rotation and samples for TA, dissolved inorganic carbon (DIC), nutrients and cell abundance were taken. Thereupon, the content of the bottles was gently transferred into smaller (2 L) polycarbonate bottles for the main experiment. Bottles were filled from bottom

to top using a hose and a funnel to limit gas exchange and avoid bubble formation. The headspace inside the bottles was kept at < 5 mL. Bottles were regularly gently rotated to keep the cultures in suspension. The growth period during the main experiment was estimated from the respective cell concentrations determined during the acclimation phase. Cells were harvested at low biomass to keep the change in dissolved inorganic carbon (DIC) due to the buildup of biogenic carbon below 4% over the growth period. Samples for cell concentrations, particulate organic (POC) and inorganic carbon (PIC) content,

and nutrient concentrations were taken at the end of the experiment.

**2.2 Determination of the carbonate system**

Samples for TA and DIC measurements were 0.2 µm filtered with low pressure, poisoned with a saturated HgCl$_2$ solution (0.5 ‰ final concentration) and stored at 4°C until analysis. TA concentrations were determined by potentiometric titration with a Metrohm Compact Titrosampler 862 and corrected with certified reference material (A. Dickson, Scripps Institution of

Oceanography, La Jolla, California; Dickson et al., 2003). DIC concentrations were measured with an Automated Infrared Inorganic Carbon Analyzer (AIRICA) equipped with a LICOR (LI-7000 CO2/H2O analyzer) detector and corrected with the certified reference material. The carbonate system parameters were calculated from salinity, temperature, TA, and DIC using the program CO2SYS (Pierrot et al., 2006) with equilibrium constants from (Mehrbach et al., 1973), refit by (Dickson and Millero, 1987).

**2.3 Determination of nutrient and carbonate system**

0.5 mL samples were transferred into Eppendorf tubes and the cell concentrations of *E. huxleyi* were measured with a flow cytometer (Accuri C6, Becton Dickinson), equipped with a 488 nm diode laser at a flow rate of 66 µL min$^{-1}$. Cells were determined based on their red fluorescence signal (> 670 nm) and forward light scatter. Daily growth rates (µ, d$^{-1}$) were determined using Eq. (1):

$\mu = \ln(N_t/N_0)/\Delta t$ ,                                                                                                                      (1)





where N denotes the cell concentration of *E. huxleyi* and t represents the time in days.

## 2.4 Particulate inorganic (PIC) and organic carbon (POC)

Samples for total particulate carbon (TPC) and particulate organic carbon (POC) were obtained by gentle filtration of the cultures through combusted (500°C for 5 hours) glass fiber filters (0.7 µm), which were stored at -20°C until analysis. POC
filters were kept in a desiccator with fuming hydrochloric acid (37 %) for 2 hours to remove all particulate inorganic carbon (PIC). After that TPC and POC filters were dried overnight at 60°C and their carbon content was measured in an elemental analyzer (Euro EA, Eurovector). The amount of PIC was determined as the difference between TPC and POC. PIC and POC production rates were calculated by multiplication of the respective cellular content with µ. The ratio of PIC to POC is indicative of the cellular degree of calcification.

## 2.5 Statistical analyses and data fitting

The data were analysed using a nonlinear regression model. Numerous treatment levels were established without replication, aiming to provide quantitative insights for ecological modelling while preserving statistical power (Cottingham et al., 2005). The coefficients of determination ($R^2$) and p-values for the regression model are presented in the results section. The Monod equations were used to analyse the growth rate, chlorophyll *a* (Chl *a*) and calcification, relating to resource concentration (e.g.
Riebesell et al., 1993; Rost et al., 2003). The Monod function is defined as Eq. (2):

$$V = V_{max} [R/(R + k_{max})],$$

where V represents the growth rate (or POC or PIC), $V_{max}$ is the maximum growth rate (or POC or PIC) at high, non-limiting concentrations, R is the resource concentration in the environment, and k is the half-saturation constant, the resource concentration at which the growth rate is half of its maximum value. All statistics and curve-fitting were conducted in Rstudio
2022.12.0 (R packages "drc" and "ggplot2"; RStudio team, 2020).





# 3 Results

## 3.1 TA manipulation and carbonate chemistry

The addition of NaOH led to a steady TA increase among the culture bottles up to a concentration of Δ1200 µmol kg$^{-1}$ (Fig.

1a). The measured TA values at the beginning of the experiment were slightly lower than the target concentrations (Fig. 1a; Tab. 1). The deviation from target concentrations increased with increasing concentrations of NaOH added. The TA values at the end of the experiment were slightly lower in most treatments. Also, the DIC concentrations show some minor fluctuations from the beginning to the end of the experiment (Fig. 1b).

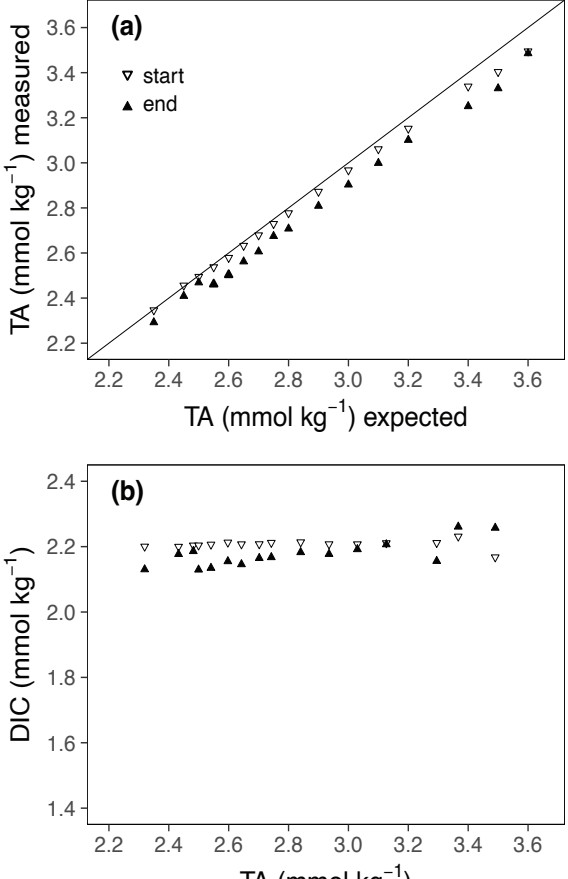

Figure 1 (a) Comparison of targeted and measured TA values at the start and end of the experiment. (b) TA:DIC diagrams at the start and end of the experiments.





The effects of increasing TA on the carbonate chemistry parameters are shown in Fig. 2 and Table 1. The concentrations of

$CO_2$ [$CO_2$], and bicarbonate ions [$HCO_3^-$] decreased with increasing TA (Fig. 2a, b), whereas the carbonate ion concentrations

[$CO_3^{2-}$] increased (Fig. 2c). The saturation state of calcite ($\Omega_{calcite}$) increases with increasing TA from 2.7 in the lowest to 24

in the highest treatment (Table 1).

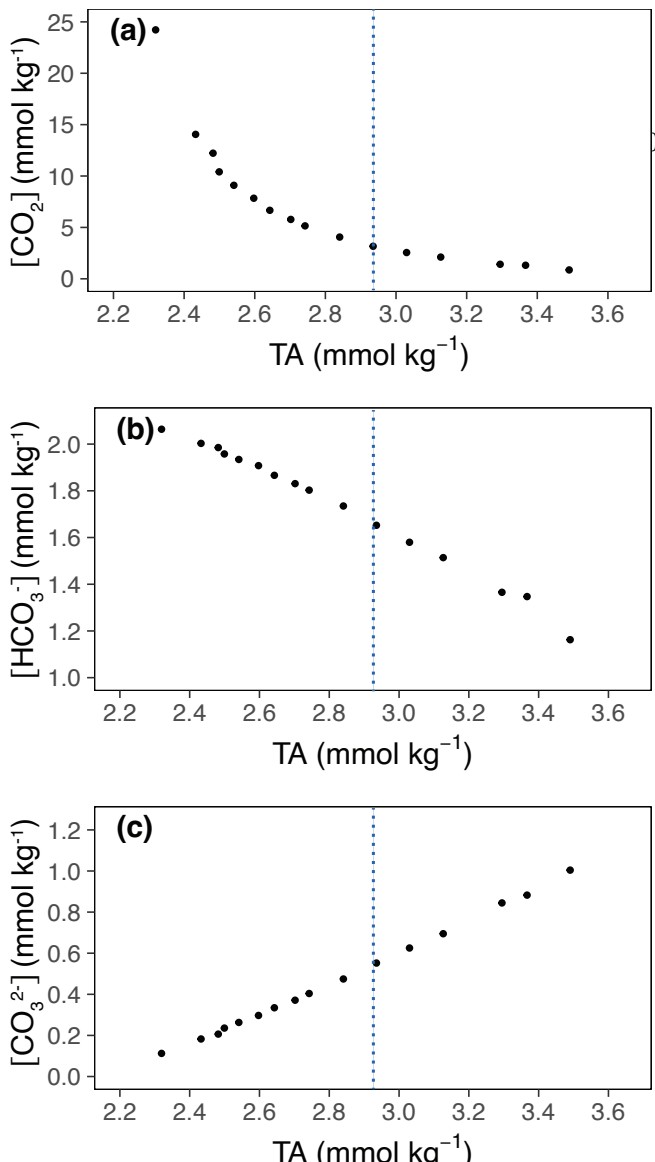

Figure 2 Relevant carbonate chemistry parameters in relation to TA. TA is the average of the values at the start and at the end of the experiment. (a) [$CO_2$]; (b) [$HCO_3^-$]; (c) [$CO_3^{2-}$]. The blue lines represent $\Delta$TA 600 µmol Kg$^{-1}$ which is the level of alkalinity beyond which *E. huxleyi* shows a decline in growth.




### 3.2 Physiological response of *E. huxleyi*

The growth rate (μ) of *E. huxleyi*, the production of POC and PIC, decreased with increasing TA (Table 2; Fig. 3a-c). There
was no significant effect of TA on the ratio of PIC to POC ($R^2 = 0.003$, $p = 0.806$) (Table 2; Fig. 3d). The growth rate declined
with decreasing $[CO_2]$ and was analyzed using the Monod equation (Fig. 4a). The fit predicts a maximum growth rate of 1.24
day$^{-1}$ (confidence interval (CI): 1.1 - 1.38 d$^{-1}$) with a residual standard error of 0.09 d$^{-1}$ (14 df), and a half-velocity constant $K_s$
$= \frac{1}{2} \mu_{max} = 2.04$ μmol $CO_2$ kg$^{-1}$. POC production was a function of μ and likewise decreased with declining $[CO_2]$ (Table 2;
Fig. 4b). The corresponding saturation function results in a maximum POC production rate of 21.10 (CI: 12.43 – 29.77) pg C
cell$^{-1}$ day$^{-1}$ with a residual standard error of 3.48 pg C cell$^{-1}$ d$^{-1}$, and a half saturation constant of 5.02 μmol kg$^{-1}$ $CO_2$. The PIC
production rate generally decreased with decreasing $[HCO_3^-]$ (Fig. 4c; $R^2 = 0.40$, $p < 0.01$) and followed the same pattern of
the saturation function plotted over the $[CO_2]$ (Fig. 4d). The Monod equation predicts a maximum PIC production rate of 9.7
pg C cell$^{-1}$ d$^{-1}$ (CI: 5.89 – 13.52) pg C cell$^{-1}$ day$^{-1}$ with a residual standard error of 2.58 pg C cell$^{-1}$ d$^{-1}$.

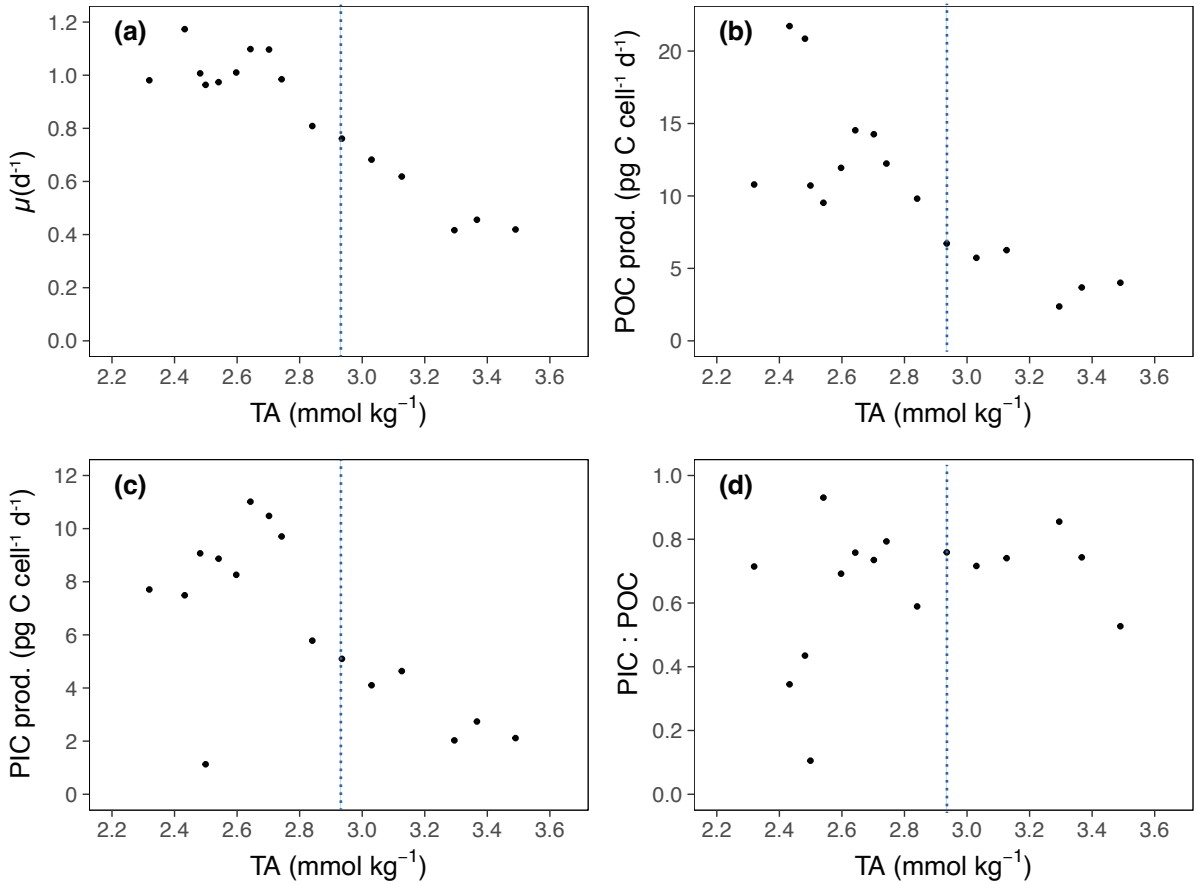

Figure 3 Physiological response parameters for *Emiliania huxleyi* in relation to TA. TA is the average of the values at the
start and at the end of the experiment. (a) Growth rates (μ). (b) particulate organic carbon production (POC prod.). (c)
particulate inorganic carbon production (PIC prod.). (d) PIC:POC ratio. The blue lines represent ΔTA 600 μmol Kg$^{-1}$ which
is the level of alkalinity beyond which *E. huxleyi* shows a decline in growth.



**4 Discussion**

This experiment was conducted according to guidelines of best practice in Ocean Alkalinity research (Oschlies et al., 2023) that emphasise the importance of small-scale laboratory experiments (Iglesias-Rodríguez et al., 2023) to contribute to assess an environmental safety approach of OAE. Bottle experiments are considered essential to comprehend the responses of keystone species to various OAE application scenarios, enabling the understanding of the physiological performance of model species and the identification of potential tipping points in OAE.

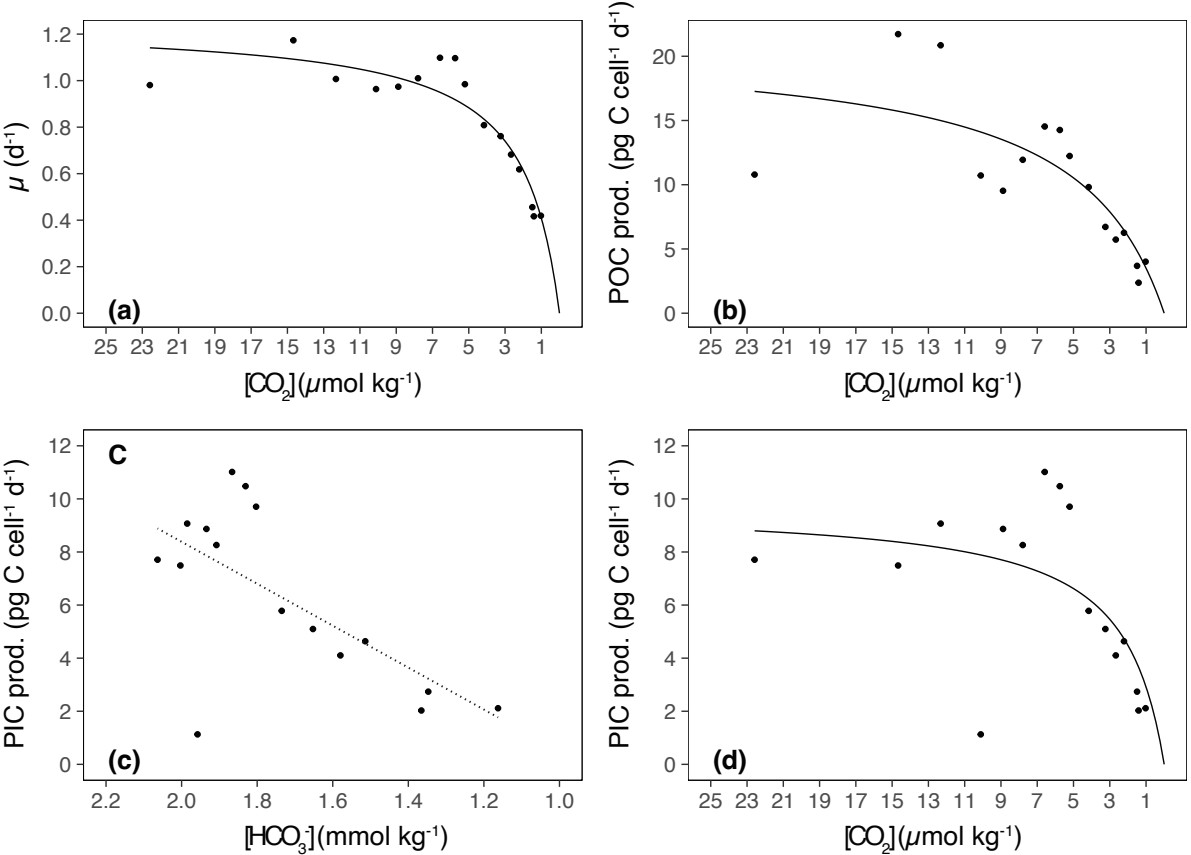

Figure 4 Growth rates, cellular POC and PIC productions in relation to $[CO_2]$ and $[HCO_3^-]$. (a) growth rate ($\mu$) in relation to $[CO_2]$. The solid line represents the Michaelis-Menten fit in the form of V = Vmax * fCO$_2$/ Kmax + fCO$_2$ where V is the growth at a specific fCO$_2$, Vmax is the theoretical maximum of growth or production, Km is the fCO$_2$ at which the maximum is half saturated ($R^2 = 0.87$, $p < 0.0001$). (b) POC production (POC prod.) in relation to $[CO_2]$, using the Michaelis-Menten fit (in this case V is the POC at a specific fCO$_2$, Vmax is the theoretical maximum of production, Km is the fCO$_2$ at which the maximum is half saturated; ($R^2 = 0.65$, $p < 0.05$). (c) PIC production (PIC prod.) in relation to $[HCO_3^-]$. The fit according to the linear regression is given ($R^2 = 0.40$, $p < 0.01$). (d) PIC production (PIC prod.) in relation to $[CO_2]$, using the Michaelis-Menten fit (in this case V is the PIC at a specific fCO$_2$, Vmax is the theoretical maximum of production, Km is the fCO$_2$ at which the maximum is half saturated; ($R^2 = 0.40$, $p = 0.12$). The biological response data are plotted against the means of the initial and final $[CO_2]$ and $[HCO_3^-]$ values.



Among the different sources of alkalinity, the addition of NaOH in a non-equilibrated way increasingly gains attention due to its smaller environmental footprint and high dissolution rates (Hartmann et al., 2023; Iglesias-Rodríguez et al., 2023; Riebesell et al., 2023). The addition of alkalinity leads to a perturbation of the carbonate chemistry, the extent of which depends on the amount of alkalinity added.

In this study, the response of the biogeochemically important coccolithophore species, *E. huxleyi,* to increasing alkalinity was tested in a system, where the carbonate chemistry was changed towards the decline of the [$CO_2$] while the total concentration of DIC remained constant. Any gas exchange and equilibration with the atmosphere were excluded, simulating the pronounced carbonate chemistry perturbations that occur in the non-equilibrated transient state immediately after alkalinity addition.

Previous studies showed that the growth rate and POC production of coccolithophores are dependent on $CO_2$ concentrations

(e.g. Krug et al., 2011; Bach et al., 2011; 2013; 2015). Most of these studies provided limited data for conditions where pH is higher than 8.4 and [$CO_2$] < 6 µmol kg$^{-1}$. These values are not unrealistic, as in natural environments, *E. huxleyi* can occasionally experience $CO_2$ limitation in post-bloom phases when [$CO_2$] falls below 7.5 µmol kg$^{-1}$ (Bach et al., 2013). Similarly, some OAE scenarios predict even stronger $CO_2$ limitation with values below 5 µmol kg$^{-1}$ (Hartmann et al., 2023; Suitner et al., 2023).

In our study, *E. huxleyi* exhibited strong reductions in growth and production of POC (below 7 pg cell cell$^{-1}$ d$^{-1}$) starting at ~ΔTA600 µmol kg$^{-1}$ (fCO$_2$ < 86 µatm; [$CO_2$] < 3.1 µmol kg$^{-1}$), suggesting a threshold below which *E. huxleyi* primary production was limited by [$CO_2$]. However, there was only a weak relationship between calcification rates and HCO$_3^-$ concentration. Furthermore, the PIC:POC ratio remained stable with values always lower than 1.

Our study doesn't have the purpose of conceptually understanding the acquisition of DIC and its subsequent use in

photosynthesis and calcification under the perturbation induced by OAE. However, the knowledge gained from ocean acidification studies can help link some physiological mechanisms when *E. huxleyi* is subjected to low [$CO_2$]. Under low [$CO_2$], the phytoplankton, to overcome the constraints inflicted by an inefficient carbon uptake and to maintain higher photosynthetic efficiency, have developed the so-called carbon concentration mechanisms (CCMs) that function to enhance the HCO$_3^-$ concentration inside the cell and promote its conversion into $CO_2$ (Badger et al., 1998). Bach et al., (2013) stated

that CCM might have evolved to help *E. huxleyi* deal with low rather than high $CO_2$ levels. Indeed, several studies indicate that $CO_2$ is the primary source of growth and photosynthesis and *E. huxleyi* upregulates its CCMs when $CO_2$ decreases to avoid a shortage in $CO_2$ (Rost et al., 2003; Bach et al., 2013). HCO$_3^-$ is an additional inorganic carbon source for growth and Chl *a* production of *E. huxleyi* (Paasche, 1964; Rost et al., 2003; Schulz et al., 2007) under low [$CO_2$]. The system demonstrates constrained effectiveness, primarily due to elevated $CO_2$ loss rates from leakage (Rost et al., 2006): observations indicate that

in coccolithophores, $CO_2$ leakage escalates as [$CO_2$] levels fall below 20 PA (Trimborn et al., 2006) concomitant with a pronounced decline in growth rates and POC production. This pattern could explain the reduction in growth and POC observed in our study when [$CO_2$] is lower than 100 µmol kg$^{-1}$ (ΔTA 600).

It is also possible that the insufficient $CO_2$ supply for growth and photosynthesis indirectly limits calcification. In principle the alkalinity addition, increased $\Omega_{calcite}$ (Table 2) which should be a favourable condition that reduces the metabolic costs for



calcification in coccolithophores (Bach et al., 2019). Our experiment shows a decrease in calcification with increasing alkalinity but the data cannot prove weather calcification is indirectly limited by the insufficient $CO_2$ and/or $HCO_3^-$ supply for photosynthesis and growth. Since it is known that the formation of the coccosphere in *E. huxleyi* is closely related to the growth of the individual cell (Kottmeier et al., 2020), we speculate that the observed reduction in PIC in our data can be explained by the the reduction in growth rate under a $[CO_2]$ limited scenario.

Gately et al. (2023), recently investigated the effect of "Moderate" ($\Delta TA \sim 700$ µmol kg$^{-1}$) and "High" ($\Delta TA \sim 2700$ µmol kg$^{-1}$) alkalinity enhancement on the coccolithophore *E. huxleyi* in comparison to the diatom *Chaetoceros* sp. The manipulation allowed atmospheric equilibration of $CO_2$ after TA manipulation and only mild fluctuations in pH were tested. The study evidenced that an equilibrated limestone-inspired alkalinity enhancement has little effect on the physiological performance of *E. huxleyi* suggesting that calcifiers may be relatively resilient to OAE when $CO_2$ is not a limiting factor. On the other hand,

our study evidenced that the increase in alkalinity without or incomplete $CO_2$ equilibration affects the growth rate of *E. huxleyi* due to $CO_2$ limitation beyond a TA increase of 600 µmol kg$^{-1}$ seawater. Thus, within the period after the addition of alkaline materials when the $CO_2$ is not yet equilibrated with the atmosphere, *E. huxleyi* may experience a competitive disadvantage compared to other species or phytoplankton groups that have more effective CCM strategies. Since the cellular calcification (PIC:POC) remains stable under increasing TA, the overall calcite production will decrease along with decreasing growth rates

in this scenario.

To conclude, to consider the applicability of OAE at a large scale, it is important to investigate the potential effects of enhanced alkalinity on both, entire marine communities as well as relevant key taxa since even small effects on single species could alter phytoplankton community composition with significant impacts on their population size that translates into their ecological and biogeochemical relevance. Given that numerous studies on ocean acidification have demonstrated species- and even strain-

specific reactions to changes in carbonate chemistry (e.g., (Langer et al., 2006) the complexity of this pattern is further emphasized. Hence further physiological data are urgently required to indicate the relevant processes that need to be investigated in community-level studies before OAE application can be considered.

**Acknowledgement**

We thank Kerstin Nachtigall, Levka Hansen and Jannes Hoffman for supporting particulate carbon measurements.

**Conflict of interest**

Allanah J. Paul has been employed by the non-profit organisation Bellona as a CDR Research and Technology Advisor since October 2023. The research reported in this manuscript was completed prior to starting this role. Allanah is also an external

scientific advisor to "Seafields" (https://www.seafields.eco/), an aquaculture business for CDR using seaweed.



**Author contribution:** MH designed the experiments with CAUT. CAUT carried out the experiment. GF analysed the data and prepared the manuscript with contributions from all co-authors.

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



| mean TA | mean DIC | mean pH | mean $fCO_2$ | $CO_2$ mean | HCO3$^-$ | $CO_3^{2-}$ | omega ar | omega ca |
|---|---|---|---|---|---|---|---|---|
| 2319 | 2165 | 7.97 | 599.49 | 22.58 | 2026 | 116 | 1.79 | 2.79 |
| 2433 | 2188 | 8.15 | 389.46 | 14.67 | 1999 | 174 | 2.68 | 4.18 |
| 2482 | 2195 | 8.22 | 327.17 | 12.32 | 1979 | 203 | 3.13 | 4.88 |
| 2499 | 2167 | 8.30 | 268.47 | 10.11 | 1923 | 234 | 3.60 | 5.61 |
| 2541 | 2171 | 8.35 | 235.88 | 8.89 | 1901 | 260 | 4.01 | 6.25 |
| 2597 | 2184 | 8.40 | 207.01 | 7.80 | 1885 | 291 | 4.48 | 6.99 |
| 2643 | 2177 | 8.46 | 174.85 | 6.59 | 1841 | 329 | 5.07 | 7.90 |
| 2702 | 2186 | 8.52 | 152.72 | 5.75 | 1814 | 366 | 5.63 | 8.79 |
| 2742 | 2190 | 8.55 | 138.38 | 5.21 | 1791 | 394 | 6.06 | 9.45 |
| 2840 | 2198 | 8.64 | 110.50 | 4.16 | 1733 | 461 | 7.10 | 11.07 |
| 2935 | 2192 | 8.73 | 86.06 | 3.24 | 1651 | 538 | 8.28 | 12.91 |
| 3030 | 2200 | 8.79 | 70.82 | 2.67 | 1590 | 607 | 9.33 | 14.56 |
| 3127 | 2209 | 8.86 | 58.84 | 2.22 | 1530 | 676 | 10.40 | 16.23 |
| 3295 | 2184 | 9.00 | 37.48 | 1.41 | 1353 | 830 | 12.76 | 19.91 |
| 3367 | 2246 | 8.99 | 39.70 | 1.50 | 1403 | 842 | 12.95 | 20.20 |
| 3490 | 2213 | 9.10 | 27.19 | 1.02 | 1244 | 967 | 14.88 | 23.21 |

**Table 1. Carbonate chemistry speciation.** The values represent the mean of the measurements taken at the beginning and at the end of the experiment. TA, DIC, HCO3$^-$, CO3$^{2-}$ are given in μmol kg$^{-1}$; fCO$_2$ in μatm; Ω aragonite and calcite are dimensionless.



420

425

430

| mean TA | Growth rate (μ) | PIC/POC | TPC production | PIC production | POC production |
|---------|-----------------|---------|----------------|----------------|----------------|
| 2319 | 0.98 | 0.71 | 18.50 | 7.71 | 10.79 |
| 2433 | 1.17 | 0.34 | 29.21 | 7.49 | 21.72 |
| 2482 | 1.01 | 0.43 | 29.92 | 9.07 | 20.85 |
| 2499 | 0.96 | 0.11 | 11.85 | 1.13 | 10.72 |
| 2541 | 0.97 | 0.93 | 18.40 | 8.87 | 9.53 |
| 2597 | 1.01 | 0.69 | 20.20 | 8.26 | 11.94 |
| 2643 | 1.10 | 0.76 | 25.55 | 11.01 | 14.53 |
| 2702 | 1.10 | 0.73 | 24.74 | 10.48 | 14.26 |
| 2742 | 0.98 | 0.79 | 21.94 | 9.71 | 12.24 |
| 2840 | 0.81 | 0.59 | 15.60 | 5.78 | 9.81 |
| 2935 | 0.76 | 0.76 | 11.81 | 5.10 | 6.72 |
| 3030 | 0.68 | 0.72 | 9.83 | 4.10 | 5.73 |
| 3127 | 0.62 | 0.74 | 10.90 | 4.64 | 6.26 |
| 3295 | 0.42 | 0.86 | 4.40 | 2.03 | 2.37 |
| 3367 | 0.46 | 0.74 | 6.42 | 2.74 | 3.68 |
| 3490 | 0.42 | 0.53 | 6.12 | 2.11 | 4.01 |

**Table 2 Cellular element quotas and production**. Growth rate (μ) are given as $d^{-1}$),
PIC:POC is dimensionless; TPC, POC and PIC productions are given as pg cell$^{-1}$*d$^{-1}$.