# Peer review of "Growth response of *Emiliania huxleyi* to ocean alkalinity enhancement"

_EGUsphere, 2024_

## Referee Comment (RC1)

**General comments**

*Overall quality of the preprint*

The manuscript by Faucher et al. is an interesting and well written study on how the globally important pelagic calcifying algae, *Emiliania huxleyi*, will respond to OAE with NaOH in lab experiments. This is an important study to understand the physiological response of individual and critical taxa to increased alkalinity. Many studies have examined the response of *E. huxleyi* to ocean acidification, so it is a great model species to study the response to alkalinity enhancement also.

One of the major findings of all the ocean acidification studies on *E. huxleyi* has been a strain specific response to acidification (e.g., Langer et al., 2009 Strain-specific responses of Emiliania huxleyi to changing seawater carbonate chemistry, Biogeosciences, 6, 2637–2646). Could it be possible that similar findings could be observed for OAE? This study is only using 1 strain of *E. huxleyi*, so perhaps using a number of other strains might provide different responses? I think it would be good if the authors can make a comment about this, and also in the discussion of the manuscript.

**Specific comments**

Line 45: do they mean ocean alkalinity or acidification studies?

Line 87: State this is a calcifying strain of *E. huxleyi*, and where the strain was obtained from (ei.e. culture collection)

Line 106-107: was there a change in growth rate during the acclimation period? Especially given that growth rate showed a decrease with TA increase. Was this a gradual decline over the acclimation time or an immediate reduction?

Line 108: define what is meant by "low" biomass (i.e. did you target a particular cell abundance or part of the growth curve?)

Line 133: what cellular concentration of PIC and POC did you use? There can be some variability between strains of *E. huxleyi* (e.g., see Harvey et al., 2015. Consequences of strain variability and calcification in Emiliania huxleyi on microzooplankton grazing; Daniels et al., 2014, Biogeochemical implications of comparative growth rates of Emiliania huxleyi and Coccolithus species). Did you use an average or the values specific for this strain?

Line 143: what is R in this context, the level of alkalinity?

Line 168: Is PIC not also a function of u?

Figure 3c and 3d: what is driving the low PIC production rate (and hence PIC:POC ratio) at 2.5 mmol kg-1 of TA? It stands out as quite the outlier.

**Technical corrections**

Line 88-89: second reference to *E. huxleyi* so can abbreviate.

---

## Referee Comment (RC2)

**Review – Faucher et al. Growth response of *Emiliania huxleyi* to ocean alkalinity enhancement**

The manuscript of Faucher et al. investigates the response of the coccolithophore *Emiliania huxelyi* (strain B92/11) to increased alkalinity (and associated decreased $CO_2$ and $HCO_3^-$ concentrations) in a culture experiment. The motivation for this study is very timely and important, given that the urgency of the climate crisis is fuelling intense focus on emissions mitigation technologies. One such approach is Ocean Alkalinity Enrichment, which may also mitigate ocean acidification. As these OAE experiments are already in trial phases, it is important to build an evidence base for the biological impacts of alkalinity enrichment on key marine organisms. The coccolithophore species *E. huxleyi* is often used as a model species for calcifying phytoplankton and is a common species in some of the regions where OAE trials have already been proposed or undertaken, e.g., in the North Atlantic shelf seas, and is therefore a very suitable candidate for the study objectives.

The data presented by Faucher et al. specifically address alkalinity perturbations rather than manipulation of other aspects of the carbonate system, which have been the focus of previous work. They identify a growth rate response to alkalinity enrichment that leads to decrease POC and PIC production, identifying a threshold value above which the response is greatest. The methods, resultant data and the manuscript are of high quality and suitable for publication following minor revisions. I have made some general suggestions for additional details that can be included to provide more context for the magnitude and rate of alkalinity enhancement that may be realistic in OAE experiments. I also think that the discussion of the manuscript would benefit from a section discussing in a bit more detail the implications of the results for guiding recommendations for alkalinity enhancement trials and/or monitoring the impacts of OAE on the biosphere and/or highlighting future research priorities.

**General comments:**

The Introduction gives a good overview of previous studies that have investigated the response of phytoplankton and coccolithophores specifically to changes in carbonate chemistry conditions. But given the wide range of carbonate chemistry parameters that have been manipulated across published studies, I think it would be helpful in the introduction to very explicitly state what (if any) impact an increase in alkalinity has on pH, $pCO_2$, DIC, and saturation state, in addition to ion concentration, so that it is very clear to the reader why pCO2-only experiments or pH-only experiments do not capture the same response as an OAE scenario.

The experiment is designed to inhibit equilibration of the culture media with the atmosphere after NaOH addition. In the field, how quickly would the surface ocean equilibrate with the atmosphere following OAE? I.e., what is a likely duration of time for the phytoplankton community to be exposed to increased alkalinity conditions, such as imposed in your experiment? Would there be a progressive decrease in alkalinity following OAE such that populations would be exposed to a gradient of alkalinity over a period of time? Or would phytoplankton populations be more likely to experience an abrupt and large increase in alkalinity that progressively equilibrates over some time? Whilst the specifics of these real-world OAE test settings are beyond the scope of your experimental study (and the details may not be available for industry reasons), some brief context of the reality of (proposed) OAE in terms of timescales and magnitude of alkalinity change (prior to the discussion where it is mentioned generally in Lns 185-189 but without specifics) would be useful context for your choice of experimental conditions and scaling from your results to real-world applications.

You mention in the Introduction that calcification consumes alkalinity but is also a $CO_2$ source, thereby offsetting $CO_2$ drawdown through OAE (Lns 51-52). However, you don't mention this process further in the discussion. Is it not relevant at the cell concentrations you are using (although this might therefore mean it is not relevant outside of the lab either as cell concentrations in the 'real world' are also low much of the time), because you have no headspace equilibration, or because the PIC:POC in your experiments in <1? Perhaps you can loop back to this aspect in the discussion, especially as calcification as a net sink or source of $CO_2$ is a common theme in coccolithophore work.

Given the obvious need for lab experiments with different phytoplankton (and other marine organisms) to provide evidence to guide safe OAE practices (which is a motivation for the research) and monitor the biological impacts of any OAE project, I feel that the manuscript should include at least a brief section in the discussion that interprets these data in the context of what is currently planned for OAE (e.g. the magnitude of alkalinity changes, timescales involved, spatial extent of OAE treatments that are likely, geographical regions of focus and how they overlap with where *E. huxleyi* is most abundant in phytoplankton communities) so that you can make some initial recommendations. This could expand on Lns 225-230, which are currently very generalised and brief. It might be that this sort of discussion is planned for future manuscripts but interpreting the dataset beyond solely discussing the response of growth rate, POC and PIC production to elevated alkalinity would link back to your initial motivation for the experiments and make the article of greater interest to a wider audience. For example, is the threshold of 600 µmol kg$^{-1}$ addition comparable to the expected change in alkalinity likely to be used in OAE, or is that much larger or smaller than likely to be used? Would your recommendation therefore be that because we now have evidence that additions beyond 600 µmol kg-1 will alter the productivity of *E. huxleyi*, OAE trials should not exceed this? Or that changes in *E. huxleyi* productivity/abundance should be monitored during OAE trials, especially in areas like the UK shelf seas and the North Atlantic where *E. huxleyi* is a very common component of the phytoplankton community? Do your results have any relevance for OAE that uses something other than NaOH (maybe other alkali that have the same impact on carbonate chemistry), or does that require further experimentation, at least for confirmation? Are your results useful for constraining biogeochemical models that investigate the impacts of OAE on the marine system?

In your PIC:POC data, there are significantly lower values for TA conditions 2433, 2482, and 2499 but you don't discuss this in the text at all – is there a likely explanation for this?

**Specific comments:**

Ln 26: I think "adapted" should be "adopted" here?

Ln 34-35: you could specify one or two examples of which minerals you are referring to here.

Ln 40: could you contextualise "gigatonnes of carbon" with respect to typical annual emissions or oceanic carbon uptakes, or similar, for context? e.g., $CO_2$ absorption equivalent to 2% current annual anthropogenic $CO_2$ emissions, or whatever it actual is.

Ln 76: I think this conclusion that *E. huxleyi* is the 'sole coccolithophore species' not showing a neutral effect to alkalinity, whilst accurate based on their analysis, comes across as a bit too decisive considering that the study in question only synthesised a limited amount of data (5 studies I think) from 5 species. I would suggest rephrasing to something more like "Of the five coccolithophore species included in the meta-analysis of Bednaršek et al. (2024), *E. huxleyi* was the only species where calcification did not show a neutral response to increased alkalinity."

Ln 87: it would be useful to mention where this strain was isolated, in addition to the culture collection information.

Ln 95 (also Ln 79 in the Introduction): NaOH is presumably one of several options for changing alkalinity chemically (I have seen proposals using magnesium hydroxide for instance). Was there a reason that you used NaOH specifically over other options?

Ln 205: should be "CCMs", plural

Ln 205: Based on carbon isotopic composition of alkenones, there is evidence that CCMs may have evolved in coccolithophores as early as the Miocene (Bolton and Stoll 2013), i.e., significantly earlier than the first appearance of *E. huxleyi*. I would suggest replacing *E. huxleyi* with Noelaerhabdaceae and rephrasing the sentence so that the evolution of CCMs sound less like a sentient choice, e.g. something along the lines of "The timing of CCM evolution may have been a response to declining atmospheric $CO_2$ concentrations during the Neogene, thus enabling Noelaerhabdaceae like *E. huxleyi* to maintain competitive growth rates under lower $CO_2$ levels".

Ln 213-214: there is an incorrect placement of a comma and a missing comma in this sentence. It should read "In principle, the alkalinity additional increased $\Omega_{calcite}$ (Table 2), which…"

Ln 216: typo – exchange "weather" with "whether".

Ln 217-219: Can you clarify what you mean here, linking coccosphere formation with growth rate and reduction in cellular PIC? With a lower growth rate, there would be on average a longer duration between successive cell divisions for the generation of new coccoliths – might this not then reasonably be expected to increase cellular PIC because there is more time for more coccoliths to be produced each cell division cycle? Or here do you refer to PIC production rather than cellular PIC?

Ln 226: can you suggest here how long this period might be – are we talking hours/days/months etc. Are there any foreseeable implications of this conclusion for when alkalinity enrichment could/should be carried out (i.e. time of year) to minimise the impact on community composition?

Ln 227: is there evidence from the literature to indicate which other phytoplankton groups, e.g., diatoms, might have a competitive advantage then in this scenario? E.g., the study of Gately et al. (2023) that also investigated *Chaetoceros*.

References: I haven't proof-read the references in any detail but can see that many need final formatting and you should check the correct referencing format for EGUsphere pre-prints.

References mentioned in this review:

Bednaršek, N., Pelletier, G., Van De Mortel, H., García-Reyes, M., Feely, R., and Dickson, A.: Unifying framework for assessing sensitivity for marine calcifiers to ocean alkalinity enhancement identifies winners, losers and biological thresholds – importance of caution with precautionary principle. EGUsphere [preprint], https://doi.org/10.5194/egusphere-2024-947, 2024.

Bolton, C. T and Stoll, H. M.: Late Miocene threshold response of marine algae to carbon dioxide limitation, Nature, 500, 558-562, https://doi.org/10.1038/nature12448, 2013.

Gately, J. A., Kim, S. M., Jin, B., Brzezinski, M. A., and Iglesias-Rodriguez, M. D.: Coccolithophores and diatoms resilient to ocean alkalinity enhancement: A glimpse of hope?, Science Advances, 9, eadg6066, https://doi.org/10.1126/sciadv.adg6066, 2023

---

## Author Comment (AC1)

We greatly appreciate the valuable comments and critical reading of the manuscript made by the two anonymous reviewers, which were useful in improving the scientific quality of the manuscript. Please find below our answers to the Reviewers' comments. For clarity, the lines mentioned in the rebuttal referred to the reviewed version of the manuscript.

Kind regards,

Giulia Faucher and co-authors

*General comments*

*Overall quality of the preprint*

*The manuscript by Faucher et al. is an interesting and well written study on how the globally important pelagic calcifying algae, Emiliania huxleyi, will respond to OAE with NaOH in lab experiments. This is an important study to understand the physiological response of individual and critical taxa to increased alkalinity. Many studies have examined the response of E. huxleyi to ocean acidification, so it is a great model species to study the response to alkalinity enhancement also.*

*One of the major findings of all the ocean acidification studies on E. huxleyi has been a strain specific response to acidification (e.g., Langer et al., 2009 Strain-specific responses of Emiliania huxleyi to changing seawater carbonate chemistry, Biogeosciences, 6, 2637–2646). Could it be possible that similar findings could be observed for OAE? This study is only using 1 strain of E. huxleyi, so perhaps using a number of other strains might provide different responses? I think it would be good if the authors can make a comment about this, and also in the discussion of the manuscript.*

In the last part of the discussion, the possible species-specific and even strain-specific responses towards OAE for coccolithophore algae were already mentioned. It also emphasised the urgency for new physiological data on the perturbation induced by OAE. Following the reviewer's suggestion, a few more references were added for clarity (line 251).

**Specific comments**

*Line 45: do they mean ocean alkalinity or acidification studies?*

If the reviewer refers to the paper by Bednaršek et al. (2024), they mean ocean acidification studies, as mentioned in the text

*Line 87: State this is a calcifying strain of E. huxleyi and where the strain was obtained from (i.e. culture collection)*

This information has been added to the text.
Line 90: *"Monospecific cultures of Emiliania huxleyi* (B92/11; Plymouth Marine Laboratory)".

*Line 106-107: was there a change in growth rate during the acclimation period? Especially given that growth rate showed a decrease with TA increase. Was this a gradual decline over the acclimation time or an immediate reduction?*

Yes, the growth rate differed already during the acclimation phase at different alkalinity levels. The cells were acclimated in all treatments for 7-9 generations to the experimental conditions. Due to varying growth rates, the acclimation period ranged from a few days to 10-12 days. This information is available in the text (Line 102).

Unfortunately, we cannot reply to the second question posed by the reviewer. Although the cell concentrations were regularly measured during the acclimation phase, we did not record daily values that allowed us to back-calculate the growth rate trend during this phase.

*Line 108: define what is meant by "low" biomass (i.e. did you target a particular cell abundance or part of the growth curve?)*

It's good practice for batch culture experiments with phytoplankton that the phytoplankton biomass at the harvest time consumes less than 5% of the total dissolved inorganic carbon. Cell concentrations were therefore kept lower

than 60.000 cells/ml. We added a sentence in the manuscript (line 90). We added a second reference to the text (Zondervan et al., 2002).

*Zondervan, I., Rost, B., and Riebesell, U.: Effect of CO2 concentration on the PIC/POC ratio in the coccolithophore Emiliania huxleyi grown under light-limiting conditions and different daylengths, Journal of Experimental Marine Biology and Ecology, 272, 55–70, https://doi.org/10.1016/S0022-0981(02)00037-0, 2002.*

*Line 133: what cellular concentration of PIC and POC did you use? There can be some variability between strains of E. huxleyi (e.g., see Harvey et al., 2015. Consequences of strain variability and calcification in Emiliania huxleyi on microzooplankton grazing; Daniels et al., 2014, Biogeochemical implications of comparative growth rates of Emiliania huxleyi and Coccolithus species). Did you use an average or the values specific for this strain?*

*If we interpret correctly the request from the reviewer, the POC and PIC production rates were calculated for each sample by multiplying growth rates with the cellular POC or PIC contents. This information is given in lines 138-134. We rephrased the text to make it more explicit.*

*"The amount of PIC was determined as the difference between TPC and POC. PIC and POC production rates were calculated for each sample by multiplying the $\mu$ with the cellular POC or PIC contents."*

*Line 143: what is R in this context, the level of alkalinity?*

R is $fCO_2$. The specifics are given in the caption of Figure 4.

*Line 168: Is PIC not also a function of u?*

If we interpret correctly the question raised by the reviewer, at line 168, POC production is defined as a function of $\mu$. The same is the case for PIC production since it is calculated by multiplying growth rates with the cellular PIC contents. As mentioned previously, it has been made more explicit in the material and method chapter.

*Figure 3c and 3d: what is driving the low PIC production rate (and hence PIC:POC ratio) at 2.5 mmol kg-1 of TA? It stands out as quite the outlier.*

We agree with the reviewer and presume the low PIC at 2.5 mmol kg$^{-1}$ of TA (specifically at 2499 µmol kg$^{-1}$) is probably an outlier. Since we haven't seen any variation from the growth rate trends and/or in the carbonate chemistry values, we presume there was a mistake during the filtration process of TPC. We decided to keep this value in the graph to give the full experiment overview (and maintain the resolution). The analyses in Figure 4 were performed considering or excluding this value and didn't change the outcome.

We added a sentence (lines 176-1179)

*"The low PIC (0.11 pg cell$^{-1}$ d$^{-1}$) value obtained at TA 2499 µmol kg$^{-1}$ is possibly an outlier. We hypothesise that there was an error in reporting the filtration me for the TPC filter for this sample. The sample was retained because no other anomalies were observed in the carbonate chemistry values or the growth of* E. huxleyi. *The statistical analyses were performed both including and excluding the sample, with no variations in the final results."*

**Technical corrections**
*Line 88-89: second reference to E. huxleyi, so can abbreviate.*

If the name of a species is at the beginning of the sentence, it should keep the full name. We didn't change the text.

---

## Author Comment (AC2)

We greatly appreciate the valuable comments and critical reading of the manuscript made by the two anonymous reviewers, which were useful in improving the scientific quality of the manuscript.

In the following sections, we have addressed each of your comments comprehensively, aiming to clarify and enhance the quality of our work as per your suggestions. For clarity, the lines mentioned in the rebuttal referred to the reviewed version of the manuscript.

Kind regards

Giulia Faucher and co-authors

Review – Faucher et al. Growth response of *Emiliania huxleyi* to ocean alkalinity enhancement

*The manuscript of Faucher et al. investigates the response of the coccolithophore Emiliania huxleyi (strain B92/11) to increased alkalinity (and associated decreased CO2 and HCO3- concentrations) in a culture experiment. The motivation for this study is very timely and important, given that the urgency of the climate crisis is fuelling intense focus on emissions mitigation technologies. One such approach is Ocean Alkalinity Enrichment, which may also mitigate ocean acidification. As these OAE experiments are already in trial phases, it is important to build an evidence base for the biological impacts of alkalinity enrichment on key marine organisms. The coccolithophore species E. huxleyi is often used as a model species for calcifying phytoplankton and is a common species in some of the regions where OAE trials have already been proposed or undertaken, e.g., in the North Atlantic shelf seas, and is therefore a very suitable candidate for the study objectives. The data presented by Faucher et al. specifically address alkalinity perturbations rather than manipulation of other aspects of the carbonate system, which have been the focus of previous work. They identify a growth rate response to alkalinity enrichment that leads to decrease POC and PIC production, identifying a threshold value above which the response is greatest. The methods, resultant data and the manuscript are of high quality and suitable for publication following minor revisions. I have made some general suggestions for additional details that can be included to provide more context for the magnitude and rate of alkalinity enhancement that may be realistic in OAE experiments. I also think that the discussion of the manuscript would benefit from a section discussing in a bit more detail the implications of the results for guiding recommendations for alkalinity enhancement trials and/or monitoring the impacts of OAE on the biosphere and/or highlighting future research priorities.*

*General comments:*

*The Introduction gives a good overview of previous studies that have investigated the response of phytoplankton and coccolithophores specifically to changes in carbonate chemistry conditions. But given the wide range of carbonate chemistry parameters that have been manipulated across published studies, I think it would be helpful in the introduction to very explicitly state what (if any) impact an increase in alkalinity has on pH, pCO2, DIC, and. saturation state, in addition to ion concentration, so that it is very clear to the reader why pCO2-only experiments or pH-only experiments do not capture the same response as an OAE scenario.*

The different approaches of OAE (i.e. equilibrated versus non-equilibrated) are mentioned in lines 65-75. The changes in the carbonate chemistry induced by a non-equilibrated OAE (lower $pCO_2$ and high pH) are mentioned in line 73. In the new version of the manuscript, following the reviewer's requests, the changes in the carbonate chemistry induced by the OAE perturbation are made more explicit. The sentence is reformulated as follows:

*"However, in a non-equilibrated OAE approach where atmospheric equilibration of $CO_2$ is prevented (sensu Hartmann et al., 2022; Suitner et al., 2023), the carbonate chemistry shifts towards lower $pCO_2$ and higher $[CO_3^{2-}]$, pH and saturation states ($\Omega$) for calcite and aragonite (Zeebe and Wolf-Gladrow, 2001). The impact of these more extreme conditions on marine calcifiers, as coccolithophores, has not yet been sufficiently investigated in order to assess the effects of ocean OAE on marine primary production."*

Reference:
Zeebe, R. E. and Wolf-Gladrow, D.: CO2 in Seawater: Equilibrium, Kinetics, Isotopes, Gulf Professional Publishing, 382 pp., 2001.

*The experiment is designed to inhibit equilibration of the culture media with the atmosphere after NaOH addition. In the field, how quickly would the surface ocean equilibrate with the atmosphere following OAE?*

*I.e., what is a likely duration of time for the phytoplankton community to be exposed to increased alkalinity conditions, such as imposed in your experiment? Would there be a progressive decrease in alkalinity following OAE such that populations would be exposed to a gradient of alkalinity over a period of time? Or would phytoplankton populations be more likely to experience an abrupt and large increase in alkalinity that progressively equilibrates over some time? Whilst the specifics of these real world OAE test settings are beyond the scope of your experimental study (and the details may not be available for industry reasons), some brief context of the reality of (proposed) OAE in terms of timescales and magnitude of alkalinity change (prior to the discussion where it is mentioned generally in Lns 185-189 but without specifics) would be useful context for your choice of experimental conditions and scaling from your results to real-world applications.*

Thanks for your comment. We added a sentence in line 190-196 to clarify and expand this concept:

*"The delivery of alkalinizing substances from platforms, pipes or ships to the Ocean is expected to cause an initial, localized impacts, potentially raising the pH above 9 (e.g., Bach et al., 2019; Suitner et al., 2023). These substances will dilute over years to decades (He and Tyka, 2023), lessening the disturbance. However, it's important to consider the biological impacts of the initial discharge. The localized, temporary increase in alkalinity and pH could create extreme conditions for marine organisms, potentially forming impact hotspots that affect phytoplankton species diversity and growth, with repercussions on trophic interactions higher up the food chain (Bach et al., 2019)."*

Reference:
Bach, L. T., Gill, S. J., Rickaby, R. E., Gore, S., and Renforth, P.: CO2 removal with enhanced weathering and ocean alkalinity enhancement: potential risks and co-benefits for marine pelagic ecosystems, Frontiers in Climate, 1, 7, 2019.

He, J. and Tyka, M. D.: Limits and CO$_2$ equilibration of near-coast alkalinity enhancement, Biogeosciences, 20, 27–43, https://doi.org/10.5194/bg-20-27-2023, 2023.

Suitner, N., Faucher, G., Lim, C., Schneider, J., Moras, C. A., Riebesell, U., and Hartmann, J.: Ocean alkalinity enhancement approaches and the predictability of runaway precipitation processes – Results of an experimental study to determine critical alkalinity ranges for safe and sustainable application scenarios, Earth System Science/Response to Global Change: Climate Change, https://doi.org/10.5194/egusphere-2023-2611, 2023.

*You mention in the Introduction that calcification consumes alkalinity but is also a CO2 source, thereby offsetting CO2 drawdown through OAE (Lns 51-52). However, you don't mention this process further in the discussion. Is it not relevant at the cell concentrations you are using (although this might therefore, mean it is not relevant outside of the lab either, as cell concentrations in the 'real world' are also low much of the time) because you have no headspace equilibration, or because the PIC:POC in your experiments in <1? Perhaps you can loop back to this aspect in the discussion, especially as calcification as a net sink or source of CO2 is a common theme in coccolithophore work.*

As mentioned in the reviewer's comment, we applied a non-equilibrated OAE manipulation ensuring that the bottles were sealed tightly to prevent CO$_2$ ingassing. Therefore, in our specific case, the increase in TA raised, in turn the pH, the of the aragonite and calcite and simultaneously caused a decline in CO$_2$. As explained in lines 229-234, this perturbation may explain the reduction in calcification due to insufficient CO$_2$ and/or HCO$_3^-$ supply. On the other hand, when OAE is applied in an equilibrated way, as discussed by Bach et al., 2019 (i.e., where the alkaline solution is pre-equilibrated with CO$_2$), calcifiers would experience more stable CO$_2$ and pH levels alongside a higher level of $\Omega$ aragonite and calcite, providing more favorable conditions for calcification. We chose not to modify the text, as we believe these concepts are already explained in the manuscript. Additionally, since our results are based on a bottle experiment involving a single strain of *Emiliania huxleyi*, we are cautious about generalizing our findings. Therefore, we decided not to expand the discussion to include the broader role of coccolithophore calcification as a net CO$_2$ sink or source under OAE scenarios.

*Given the obvious need for lab experiments with different phytoplankton (and other marine organisms) to provide evidence to guide safe OAE practices (which is a motivation for the research) and monitor the biological impacts of any OAE project, I feel that the manuscript should include at least a brief section in the discussion that interprets these data in the context of what is currently planned for OAE (e.g. the magnitude of alkalinity changes, timescales involved, spatial extent of OAE treatments that are likely, geographical regions of focus and how they overlap with where E. huxleyi is most abundant in phytoplankton communities)*

*so that you can make some initial recommendations. This could expand on Lns 225-230, which are currently very generalised and brief. It might be that this sort of discussion is planned for future manuscripts but interpreting the dataset beyond solely discussing the response of growth rate, POC and PIC production to elevated alkalinity would link back to your initial motivation for the experiments and make the article of greater interest to a wider audience. For example, is the threshold of 600 μmol kg-1 addition comparable to the expected change in alkalinity likely to be used in OAE, or is that much larger or smaller than likely to be used? Would your recommendation therefore be that because we now have evidence that additions beyond 600 μmol kg-1 will alter the productivity of E. huxleyi, OAE trials should not exceed this? Or that changes in E. huxleyi productivity/abundance should be monitored during OAE trials, especially in areas like the UK shelf seas and the North Atlantic where E. huxleyi is a very common component of the phytoplankton community? Do your results have any relevance for OAE that uses something other than NaOH (maybe other alkali that have the same impact on carbonate chemistry), or does that require further experimentation, at least for confirmation? Are your results useful for constraining biogeochemical models that investigate the impacts of OAE on the marine system?*

We thank the reviewer for this thoughtful comment. We took the opportunity to revise parts of the introduction and the discussion/conclusion sections. While we appreciate the suggestions made by the reviewer, we do not fully agree with all of them. Many of the questions raised extend far beyond the scope of this work and are too complex to be addressed by a single bottle experiment. We will clarify our position in the following lines:

1. *I feel that the manuscript should include at least a brief section in the discussion that interprets these data in the context of what is currently planned for OAE (e.g. the magnitude of alkalinity changes, timescales involved, spatial extent of OAE treatments that are likely, geographical regions of focus and how they overlap with where E. huxleyi is most abundant in phytoplankton communities) so that you can make some initial recommendations:*

We believe that laboratory experiments are essential for identifying the root cause of physiological responses, as they allow for observations in highly controlled environments using model species. Laboratory studies help establish cause-effect relationships. In the context of OAE, they are necessary for generating hypotheses to test in the field. However, drawing conclusions from a culture experiment with a single strain of one species is not appropriate, as bottle experiments, by definition, cannot provide insight into ecosystem-level impacts of OAE.

2. *For example, is the threshold of 600 μmol kg-1 addition comparable to the expected change in alkalinity likely to be used in OAE, or is that much larger or smaller than likely to be used? Would your recommendation therefore be that because we now have evidence that additions beyond 600 μmol kg-1 will alter the productivity of E. huxleyi, OAE trials should not exceed this? Or that changes in E. huxleyi productivity/abundance should be monitored during OAE trials, especially in areas like the UK shelf seas and the North Atlantic where E. huxleyi is a very common component of the phytoplankton community?*

Following the recommendations of the Best Practice Guide for OAE (Oschlies et al., 2023), this study tested the response of a species of coccolithophore algae to elevated TA conditions. The range tested remains within the proposed alkalinity targets, reaching approximately 3000–4000 $\mu$mol kg$^{-1}$ (Renforth and Henderson, 2017). In this experiment, the highest TA tested was below 4000 $\mu$mol kg$^{-1}$. While these alkalinity conditions may appear extreme, they can be valuable for identifying thresholds of alkalinity enhancement. Although these conditions might fall outside worst-case scenario projections, they contribute to a better understanding of biological responses to OAE applications. We add a sentence in the discussion (lines 196-198).

*"The range tested remains within the proposed alkalinity targets, reaching approximately 3000–4000 $\mu$mol kg$^{-1}$ (Renforth and Henderson, 2017). In this experiment, the highest TA tested was below 4000 $\mu$mol kg$^{-1}$."*

Reference:
Renforth, P. and Henderson, G.: Assessing ocean alkalinity for carbon sequestration, Reviews of Geophysics, 55, 636–674, 2017.

3. *Do your results have any relevance for OAE that uses something other than NaOH (maybe other alkali that have the same impact on carbonate chemistry), or does that require further experimentation, at least for confirmation?*

Reviewer 2 raised a specific question regarding our choice to test NaOH. We have clarified our reasoning by adding a sentence in the discussion (lines 188-189). A more comprehensive explanation is also provided in the subsequent lines of this rebuttal (see the Specific Comment section).

Among the different sources of alkalinity, the addition of NaOH in a non-equilibrated way increasingly gains attention due to its smaller environmental footprint and high dissolution rates (Hartmann et al., 2023; Iglesias-Rodríguez et al., 2023; Riebesell et al., 2023). *For these reasons, NaOH is considered one of the most suitable feedstocks for OAE in pelagic environments (Eisaman et al., 2023; Iglesias-Rodríguez et al., 2023), with field trials already considering its use."*

4. *Are your results useful for constraining biogeochemical models that investigate the impacts of OAE on the marine system?*

This study, to the best of our knowledge, is the first to evaluate the impact of non-equilibrated OAE on a model species like *E. huxleyi*, laying a solid groundwork for future research in this field. However, a single-species experiment is not enough to constrain biogeochemical models on OAE. To do so, we need to integrate different levels of biological organization from species to ecosystem.

*In your PIC:POC data, there are significantly lower values for TA conditions 2433, 2482, and 2499, but you don't discuss this in the text at all – is there a likely explanation for this?*

The low values at TA 2433 and 2492 $\mu mol\ kg^{-1}$ of the PIC:POC, depend on the high POC values obtained for these two samples. The low values of the PIC:POC at 2499 $\mu mol\ kg^{-1}$ might be the result of an error during the filtration process and, therefore a possible outlier. We added a sentence in the results on lines 179-182

*"The low PIC (0.11 pg cell-1*d-1) value obtained at TA 2499 $\mu mol\ kg^{-1}$ is likely an outlier. We hypothesized that this discrepancy may be due to an error in reporting the filtration volume for TPC for this sample. Despite this, the sample was retained for analysis, as no other anomalies were observed in either the carbonate chemistry parameters or the growth of E. huxleyi. Statistical analyses were conducted both with and without this sample, and no significant differences were observed in the final results."*

**Specific comments:**

*Ln 26: I think "adapted" should be "adopted" here?*
The text has been modified.

*Ln 34-35: you could specify one or two examples of which minerals you are referring to here.*

The text was modified, adding a few examples.

*"One proposed method to remove CO2 from the atmosphere mimics the natural process of rock weathering, whereby suitable minerals (i.e.* Olivine, Basalt, Carbonate*) are extracted and introduced into the surface ocean (Ocean Alkalinity Enhancement, OAE; Gattuso et al., 2015; GESAMP, 2019)."*

*Ln 40: could you contextualise "gigatonnes of carbon" with respect to typical annual emissions or oceanic carbon uptakes, or similar, for context? e.g., CO2 absorption equivalent to 2% current annual anthropogenic CO2 emissions, or whatever it actual is.*

To the best of our knowledge, no study so far has been able to give such an estimate.
The sentence was revised as follows:

*"Theoretical studies indicate that OAE has the potential to remove 3 to 30 gigatonnes of carbon dioxide/year (; Renforth and Henderson, 2017; Feng et al., 2017) from the atmosphere and thus contribute to global climate change mitigation efforts."*

*Ln 76: I think this conclusion that E. huxleyi is the 'sole coccolithophore species' not showing a neutral effect to alkalinity, whilst accurate based on their analysis, comes across as a bit too decisive considering that the study in question only synthesised a limited amount of data (5 studies I think) from 5 species. I would suggest rephrasing to something more like "Of the five coccolithophore species included in the meta-analysis of Bednaršek et al. (2024), E. huxleyi was the only species where calcification did not show a neutral response to increased alkalinity."*

We followed the reviewer suggestion and changed the text accordingly.

*Ln 87: it would be useful to mention where this strain was isolated in addition to the culture collection information.*

The strain was obtained from the Plymouth Marine Laboratory. The information is now added to the text.

*Ln 95 (also Ln 79 in the Introduction): NaOH is presumably one of several options for changing alkalinity chemically (I have seen proposals using magnesium hydroxide for instance). Was there a reason that you used NaOH specifically over other options?*

We agree with the reviewer that NaOH is just one of the several potential feedstocks that can be used for OAE. We choose NaOH for several reasons:

1. NaOH is one of the most suitable feedstocks for OAE approaches, with field trials already considering its use (i.e., LOC-NESS project PI: Adam Subhas; https://subhaslab.whoi.edu/loc-ness/).
2. NaOH is considered one of the most suitable feedstocks to be used in pelagic environments since it dissolved easily in seawater compared to other materials (e.g., $Mg(OH)_2$)
3. We aimed to investigate the effects of the carbonate chemistry changes induced by the addition of alkaline substances without introducing other elements (e.g., trace metals). For this reason, we discarded other potential materials and we chose NaOH.

*Ln 205: should be "CCMs", plural*
CCMs was already plural in the original version at line 206

*Ln 205: Based on carbon isotopic composition of alkenones, there is evidence that CCMs may have evolved in coccolithophores as early as the Miocene (Bolton and Stoll 2013), i.e., significantly earlier than the first appearance of E. huxleyi. I would suggest replacing E. huxleyi with Noelaerhabdaceae and rephrasing the sentence so that the evolution of CCMs sound less like a sentient choice, e.g. something along the lines of "The timing of CCM evolution may have been a response to declining atmospheric CO2 concentrations during the Neogene, thus enabling Noelaerhabdaceae like E. huxleyi to maintain competitive growth rates under lower CO2 levels"*

We followed the reviewer's advice and rephrased the sentence accordingly (now at Line 219)

Ln 213-214: there is an incorrect placement of a comma and a missing comma in this sentence. It should read "In principle, the alkalinity additional increased Wcalcite (Table 2), which…"

Thanks for the comment. The text has been modified accordingly.

*Ln 216: typo – exchange "weather" with "whether".*
Text modified

*Ln 217-219: Can you clarify what you mean here, linking coccosphere formation with growth rate and reduction in cellular PIC? With a lower growth rate, there would be on average a longer duration between successive cell divisions for the generation of new coccoliths – might this not then reasonably be expected to increase cellular PIC because there is more time for more coccoliths to be produced each cell division cycle? Or here do you refer to PIC production rather than cellular PIC?*

Yes, we refer to PIC production, and we added "production" to the text for clarity.
As mentioned in the text, Kottmeier et al.(2020) demonstrated that the cellular ratio of calcium carbonate to organic carbon remains relatively stable throughout the cell cycle. The production of calcium carbonate is closely linked with the increase in biomass and volume of exponential growth (Kottmeier et al., 2020). Therefore, the

number of coccoliths remains fairly constant in proportion to the cell volume throughout the cell cycle. In other words, newly divided cells are small but fully covered with coccoliths.

Reference:
*Kottmeier, D. M., Terbrüggen, A., Wolf-Gladrow, D. A., and Thoms, S.: Diel variations in cell division and biomass production of Emiliania huxleyi—Consequences for the calculation of physiological cell parameters, Limnology and Oceanography, 65, 1781–1800, https://doi.org/10.1002/lno.11418, 2020.*

**Ln 226: can you suggest here how long this period might be – are we talking hours/days/months etc. Are there any foreseeable implications of this conclusion for when alkalinity enrichment could/should be carried out (i.e. time of year) to minimise the impact on community composition?**

1. We added this information to the text along with a reference (line 243): *"CO$_2$ equilibration would occur on the order of weeks to months (Ringham et al., 2024), thus, within the period after the addition of alkaline* substances, *E. huxleyi may experience a competitive disadvantage compared to other* species or phytoplankton groups that have more effective CCM strategies."
2. Regarding the second request, as mentioned in the text, the strain of *E. huxleyi* used for this experiment suggests that for non-equilibrated OAE to be applied safely, the ΔTA should not exceed 600 μmol kg⁻¹. However, we are cautious about extrapolating broader conclusions on when OAE should be applied to minimize its impact on community composition, as our findings are based on a single-bottle experiment with one strain of *E. huxleyi*.

*Ln 227: is there evidence from the literature to indicate which other phytoplankton groups, e.g., diatoms, might have a competitive advantage then in this scenario? E.g., the study of Gately et al. (2023) that also investigated Chaetoceros.*

We added to the sentence a few references of studies that investigated the carbon acquisition of marine phytoplankton. These studies were aimed at understanding the response of marine phytoplankton to OA. Through testing several CO$_2$ ranges, they evidenced a more efficient CCM of diatoms and dinoflagellates compared to coccolithophore and specifically to *E. huxleyi*.

In the context of OAE, to the best of our knowledge, only a few studies investigate the response of phytoplankton to non-equilibrated OAE conditions in culture experiments. Gately et al. (2023), as mentioned before, tested a different OAE scenario where TA was manipulated in an equilibrated/semi-equilibrated way. Therefore, the pH only slightly increased, and the algae did not experience CO$_2$ limitation conditions. In a study still under discussion by Oberlander et al., (Biogeosciences, under discussion), the authors investigated the impacts of short-term elevation in pH after OAE manipulation on the growth rates of a diatom and a prymnesiophyte. This study shows no significant impacts on the growth rates of the diatom *Thalassiosira pseudonana* and the prymnesiophyte *Diacronema lutheri* with short-term (10-minute) exposure to elevated pH but evidenced a significant decrease in growth rates with long-term (8 days). The study by Oberlander et al. (2024) has not been discussed in our study because it focuses on identifying pH threshold values rather than implications for CO$_2$ limitation conditions.

Reference:
Oberlander, J. L., Burke, M. E., London, C. A., & MacIntyre, H. L. (2024). Assessing the impacts of simulated Ocean Alkalinity Enhancement on viability and growth of near-shore species of phytoplankton. *EGUsphere*, *2024*, 1-21.

**References: I haven't proof-read the references in any detail but can see that many need final formatting and you should check the correct referencing format for EGUsphere preprints.**

*References were checked and reformatted when needed.*

References mentioned in this review:

Bednaršek, N., Pelletier, G., Van De Mortel, H., García-Reyes, M., Feely, R., and Dickson, A.: Unifying framework for assessing sensitivity for marine calcifiers to ocean alkalinity enhancement identifies winners, losers and biological thresholds – importance of caution with precautionary principle. EGUsphere [preprint], https://doi.org/10.5194/egusphere-2024- 947, 2024.

Bolton, C. T and Stoll, H. M.: Late Miocene threshold response of marine algae to carbon dioxide limitation, Nature, 500, 558-562, https://doi.org/10.1038/nature12448, 2013.

Gately, J. A., Kim, S. M., Jin, B., Brzezinski, M. A., and Iglesias-Rodriguez, M. D.: Coccolithophores and diatoms resilient to ocean alkalinity enhancement: A glimpse of hope?, Science Advances, 9, eadg6066, https://doi.org/10.1126/sciadv.adg6066, 2023